Salinity and pH effects on survival, growth, and reproduction of quagga mussels

Seitz Carina 1 2 3 cseitz@comahue-conicet.gob.ar
Scordo Facundo 1 4 5
Suenaga Erin 1
Carlson Emily M. 1
McMillen Shaye 1
Gregory Logan 1
Chandra Sudeep 1
1 Department of Biology, Global Water Center and University of Nevada, Reno , Reno, NV , United State of America
2 IPATEC, Centro Regional Universitario Bariloche (CRUB), (CONICET-UNCO) , San Carlos de Bariloche, Río Negro , Argentina
3 Departmento de Geología, Universidad Nacional del Sur , Bahía Blanca, Buenos Aires , Argentina
4 Instituto Argentino de Oceanografia (IADO), Universidad Nacional del Sur (UNS)-CONICET , Bahía Blanca, Buenos Aires , Argentina
5 Departamento de Geografía y Turismo, Universidad Nacional del Sur , Bahía Blanca, Buenos Aires , Argentina
Oehlmann Jörg
Electronic publication date: 2023 May 26
Publication date: 2023
Volume: 11
Electronic Location ID: e15450
Received 2022 Nov 9; Accepted 2023 May 3
Copyright: © 2023 Seitz et al.
Copyright year: 2023
Copyright holder: Seitz et al.
License: This is an open access article distributed under the terms of the Creative Commons Attribution License, which permits unrestricted use, distribution, reproduction and adaptation in any medium and for any purpose provided that it is properly attributed. For attribution, the original author(s), title, publication source (PeerJ) and either DOI or URL of the article must be cited.
License URL: https://creativecommons.org/licenses/by/4.0/

Keywords: Invasive species, Quagga mussel, Dreissena spp, Freshwater lake, Brackish lake, Salinity, pH

Funding: Dan Mosley and the United States Department of Interior Bureau of Indian Affairs A20AV00856 Aquatic Invasive Species Program This work was supported by Dan Mosley and the United States Department of Interior Bureau of Indian Affairs under Grant No A20AV00856, Aquatic Invasive Species Program. The funders had no role in study design, data collection and analysis, or preparation of the manuscript. The funders did have a role in the decision to publish.

==============================
Background

In recent decades, invasive quagga mussels have expanded to the Western United States from the Great Lakes region of North America. Most studies that evaluate the invasion potential of quagga mussels in western water bodies have utilized physiological and life history information from zebra mussels, a related taxon. Few studies have assessed the potential for invasion using specific information from quagga mussel life history or experiments that test for their survival in the fresh and saline waters of the western United States.

Methods

We investigated quagga mussel survival, growth, and reproduction using semi-natural experiments under temperature and light controlled conditions across a gradient of water salinity (fresh to brackish) and pH (8.4–11). Water from Lake Mead was used as a positive control in our experiment, and water from Pyramid Lake and the Truckee River was used as brackish and freshwater treatments, respectively. The mussels used in the experiments were collected from Lake Mead.

Results

After 12 h in brackish water (4 ppt, pH 9.3), we observed 100% mortality of adult mussels. The swelling and disintegration of body tissues and high mortality rates indicated that high potassium, sodium, and chloride concentrations were the likely causes of death in brackish water treatments. In contrast, mussels were able to survive, grow, and reach sexual maturity in freshwater (0.1 ppt) with a low calcium concentration (17 mg L−1) after 57 days. Mussels died after 2 days at pH 11 and after 12 days at pH 10; during the 14-day monitoring period, no mortality was detected at pH 9.0, 9.3, or 9.5 and mussels did not exhibit any visual indications of stress. Understanding quagga mussel physiological and environmental tolerances appears to be essential for assessing their invasion potential in aquatic habitats.

Introduction

The proliferation of exotic species has been exacerbated by the globalization of economies and trade (Karatayev et al., 2007). In the Northern Hemisphere, invasive zebra (Dreissena polymorpha (Pallas, 1771)) and quagga (Dreissena rostriformis bugensis (Andrusov, 1897)) mussels are one of the most aggressive freshwater invaders and have enormous economic and ecological impacts (Karatayev, Burlakova & Padilla, 2015). These species alter the ecology of lakes and rivers by decoupling pelagic and benthic trophic pathways, increasing offshore clarity, stimulating benthic production, and altering biodiversity (Ricciardi, Neves & Rasmussen, 1998; Makarewicz, Lewis & Bertram, 1999; Bially & MacIsaac, 2000). Mollusks, including zebra and quagga mussels, are the fifth most costly invasive species in the United States with an annual cost estimate of $4.8 billion (Fantle-Lepczyk et al., 2022). The cost is almost five times higher than was predicted in 2005 for damages and associated control expenses in the United States (Pimentel, Zuniga & Morrison, 2005). Invasive mussel species can be easily transported to new water bodies, however, the physiochemical characteristics necessary for successful establishment are still poorly understood.

The two mussel species colonized the mid-western and eastern regions of North America around the same time; zebra mussels were discovered in the Great Lakes region in 1986 (Hebert, Muncaster & Mackie, 1989) and quagga mussels were found in Lake Erie in 1989. However, the quagga mussel was not recognized as a distinct species until 1991 (May & Mardsen, 1992; Mills et al., 1996). In addition to invading the Great Lakes, both Dreissena spp. had colonized a total of 772 water bodies in the United States and Canada by 2010, with zebra mussels occupying 17 times more water bodies than quagga mussels (Benson, 2014). In the early 2000s, a shift was observed and quagga mussels became the predominant first colonizers, favored by small differences in physiology and depth of reproduction (Ram et al., 2012). Quagga mussels first appeared in the western U.S. in Lake Mead (AZ-NV USA) in early 2007 (Stokstad, 2007) and rapidly spread across the Colorado River system colonizing 30 lakes and reservoirs in Arizona, California, and Nevada by the end of 2008 (Nalepa, 2010). Recently, it was observed that zebra mussels have infested Highline Lake in the state of Colorado (Colorado Parks & Wildlife, 2023). These recent invasions have spurred efforts to determine the invasion risk posed by zebra and quagga mussels in inland waters of the western United States.

Although there has been an increase in the number of publications about Dreissena spp. since their first appearance in North America, only 30% of the articles are related to quagga mussels (Passamaneck, 2018; Karatayev & Burlakova, 2022). Consequently, the information available on their life history and environmental conditions that could promote establishment may not be sufficient to predict their potential spread and the ecological implications of invasion (Nalepa, 2010; Karatayev, Burlakova & Padilla, 2015). The most commonly used variables for assessing potential mussel invasion are pH and calcium (Ca) concentrations, but quagga mussel tolerance to these variables is still poorly understood. Several studies indicated that quagga mussels may be less tolerant to low Ca concentrations (>12 mg L−1) relative to zebra mussels (>8 mg L−1) since they are usually found in environments with higher Ca concentrations (>12 mg L−1) (Zhulidov et al., 2004; Jones & Ricciardi, 2005; Baldwin et al., 2012). However, quagga mussels could establish in Lake Tahoe, which has low Ca concentrations (9 mg L−1) (Davis et al., 2015), but would require higher concentrations (>18 mg L−1) for the establishment of permanent populations (Baldwin et al., 2012). Permanent populations of adult quagga mussels in water with low Ca concentrations have been observed in places that are hydrologically connected to waters where veligers can develop (Cohen, 2007). Zebra mussel pH tolerance has also been used to evaluate the potential for quagga mussel invasion (Mackie & Claudi, 2010). Only two studies have assessed quagga mussel tolerance to different pH gradients, and they have only focused on the use of very high (pH > 12) or low pH (<7.5) to prevent or control mussel invasions (Claudi et al., 2012, 2013) (Table 1). These studies did not have conclusive results regarding the upper pH limits tolerated by quagga mussels due to technical issues during the experiments (Claudi et al., 2013). Similar to pH, upper limits for salinity have been more intensively examined for zebra mussels over quagga mussels. Salinity tolerances determined by laboratory experiments and presence-absence data are highly variable, but suggest that adults can tolerate up to 6–8 ppt when first acclimated to a lower salinity concentration (Garton, McMahon & Stoeckmann, 2014). Quagga mussels are considered a euryhaline species, living in both freshwater and brackish water in their native ranges (Orlova et al., 2005), and have shown rapid adaptation to new environmental conditions after only a few generations (Mills et al., 1996). Studies of Dresseina spp. salinity tolerance have not been done since the mid-1990s, so analyzing the salinity tolerance of the new generations of Dreissena spp. in North America are warranted despite their absence from brackish waters to date.

Table 1 Water quality variables of the water treatment from field locations stored in the tanks and containers holding the mussels.

Parameter	Lake mead	Popcorn rock	Warrior point	Truckee river	
Chalk variables	
Calcium (mg L−1 Ca)	54.8 ± 6.0	8.8 ± 1.0	8.2 ± 0.8	16 ± 1.2	
pH	8.3 ± 0.1	9.3 ± 0.1	9.3 ± 0.1	8.1 ± 0.1	
Total alkalinity
(mg L−1 CaCO3)	135.0 ± 6.0	1012.0 ± 25.0	1068.0 ± 25.0	86.0 ± 8.0	
Bicarbonate alkalinity
(mg L−1 CaCO3)	135.0 ± 6.0	793.0 ± 36.0	800.0 ± 69.0	86.0 ± 8.0	
Carbonate alkalinity
(mg L−1 CaCO3)	-	219.0 ± 21.0	263.0 ± 55.0	-	
Total hardness (mg L−1 CaCO3)	278.0 ± 17.0	455.0 ± 8.0	475.0 ± 14.0	83.5 ± 8.0	
Nutrient variables	
Dissolved oxygen (%)	82.5 ± 2.6	77.4 ± 0.4	74.5 ± 0.5	81.9 ± 4.0	
Chlorophyll a (µg L−1)	0.1 ± 0.1	0.8 ± 1.1	0.9 ± 1.3	0.1 ± 0.1	
Total phosphorus (µg L−1)	11.0 ± 4.0	206.0 ± 15.0	114.0 ± 38.0	68.2 ± 22.0	
Total nitrogen (mg L−1)	0.5 ± 0.2	0.5 ± 0.1	0.6 ± 0.1	0.5 ± 0.2	
Physical variables	
Temperature	20.6 ± 1.2	19.7 ± 0.4	19.6 ± 0.1	20.8 ± 1.4	
Conductivity (uS cm−1)	9.3 ± 0.20	75.1 ± 5.00	75.2 ± 6.00	3.6 ± 0.20	
Salinity (ppt)	0.4 ± 0.1	4.07 ± 0.1	4.34 ± 0.1	0.17 ± 0.1	
Other chemical variables	
SAR	2.4 ± 0.1	2.2 ± 0.1	2.3 ± 0.1	2.0 ± 0.1	
Na (Soluble)
(mg L−1)	87.1 ± 3.7	99.5 ± 2.1	101.8 ± 6.9	40.00 ± 3.4	
Mg (Soluble)
(mg L−1)	24.9 ± 1.1	83.8 ± 4.5	89.7 ± 13.2	7.9 ± 0.6	
B (Soluble)
(mg L−1)	0.1 ± 0.0	9.7 ± 0.3	9.8 ± 1.5	0.4 ± 0.0	
Cl (mg L−1)	87. 8 ± 3.5	1748.4 ± 61.3	1558.0 ± 442.0	40.7 ± 2.5	

We conducted site specific studies to assess the potential for adult quagga mussels to establish in fresh to saline water ecosystems, including terminal (endorheic) lakes, that are present in the western United States. Site specific studies suggested the potential for adult quagga mussel establishment in the freshwater of Lake Tahoe (Davis et al., 2015), which is connected to the more brackish waters of Pyramid Lake through the Truckee River. In this study, we investigated quagga mussel survival, growth, and reproduction in an ecosystem with a salinity (fresh to brackish) and pH (>8) gradient, as well as their tolerance to high pH concentrations. Specifically, our objectives were to (1) determine adult quagga mussel survival, growth, and potential for producing viable gametes that could result in the establishment and spread of mussels in Pyramid Lake (brackish water) and the Truckee River (fresh water); (2) explore which factors (pH, salinity, or both) are causing the adult quagga mortality in Pyramid lake water; and (3) analyze the role of pH on adult quagga mussel survival.

Study area

We implemented seminatural laboratory experiments to determine adult quagga mussel survival and growth in water from two different locations within Pyramid Lake (brackish waters of Warrior Point and Popcorn Rock) and one location in the lower basin of the Truckee River (freshwater below Marble Bluff Dam, 1.6 km above Pyramid Lake). These locations were selected to capture the salinity and pH gradients of the Truckee River-Pyramid Lake ecosystem. Currently, there are no established populations of quagga mussels at these sites. Pyramid Lake is a large (487 km2, maximum depth 105 m) terminal desert lake located within the Pyramid Lake Paiute Tribe Reservation in western Nevada, approximately 50 km northeast of Reno (Fig. 1). Pyramid Lake receives about 85% of its annual water input from the Truckee River, which originates in the upper watershed of the Lake Tahoe Basin in the Sierra Nevada Mountains (Fig. 1). As a positive control for the experiments, we utilized water from the Lake Mead Fish Hatchery which draws water from Lake Mead, the largest reservoir in the United States. Quagga mussels have established populations throughout the reservoir and Colorado River since 2007 (Fig. 1).

Figure 1 Location of the study area.

(A) Location of the study area within the United States. (B) Location of Pyramid Lake and Lake Mead in the western United States; (C) location of the study sites where water was collected for the experiments in Pyramid Lake (Warrior Point; Popcorn Rock) and the Truckee River (Marble Bluff Dam); (D) location of the study sites where water (Fish Hatchery) and quagga mussels (Lake Mead Marine) were collected for the experiment in Lake Mead; (E) experimental set-up using 13 L containers with mussels under different water treatments in a climatized room; (F) dead mussels retrieved after 19 h in brackish water with a bloated appearance and tissue disintegration; (G) mussel growth in Truckee River water treatment after 58 days.

Materials and Methods

Collection, handling of mussels, and assessment of survival in the experiments

We assessed survival and growth of adult quagga mussels of different sizes and under different water quality conditions through a series of seminatural experiments. We collected adult quagga mussels (>9 mm) from the Lake Mead Marina via a special permit provided by the US State of Nevada. The mussels were carefully removed from ropes hanging from the dock of the marina. The mussels were then transported to our laboratory in containers with Lake Mead water and bubblers to keep the water well oxygenated. We let the mussels acclimatize in Lake Mead water for 48 h at 20 °C before placing them in the different water treatments. Length and wet-weight were measured pre- and post- experiments. The length of individual mussels was measured using a digital caliper (General Ultratech®). We also measured change in wet mass, including live soft tissue and shell mass, to assess total mussel growth throughout the experiment. Individual live mussels were dried externally with a tissue and weighed on a digital scale (Mettler Toledo AB104-S®; Mettler Toledo, Columbus, OH, USA). Each day, we noted the behavior of the mussels (e.g., opening and extraction of the siphon, attachment to container surfaces, and motility) and the status of the mussels’ survival (live or dead) by noting gape and response to physical contact. Water from each ecosystem was stored near the laboratory in 2,273 L opaque tanks. We carried out the experiments in a temperature-controlled room using a 12-h day-night cycle to simulate natural conditions and optimize the filtering activity of the mussels. At the end of the experiment, the mussels were dried at 95 °C to ensure complete death before being discarded. To avoid contaminating local water systems with quagga mussels or veliger larvae, we treated all water and containers with 6% sodium hypochlorite bleach solution for 24 h before being discarded according to the Guide to Preventing Aquatic Invasive Species Transport by Wildland Fire Operations (National Wildfire Coordinating Group, 2017).

Adult mussels’ survival, growth, and reproduction in brackish to freshwater ecosystems

Experiment 1: Individual mussels of similar size (18 ± 2 mm length) were placed in 13 L containers in a climatized room at 20 °C (1 mussel per container with 30 replicates per treatment) (Fig. 1E). Treatments included water from the following locations: Lake Mead (control where mussels have established since 2007), Pyramid Lake from Popcorn Rock (brackish water), and Pyramid Lake from Warrior Point (brackish water).

Experiment 2: To determine if the size of the quagga mussel affected their survival and mortality rate, we implemented another experiment using the same water treatments as Experiment 1 but with mussels of three different sizes: large (16–26 mm length, N = 30), medium (8–16 mm length, N = 30), and small (3–8 mm length, N = 20). The individual mussels were monitored for survival hourly for 12 h. The number of replicates was lower in the small size class due to a lack of mussels within this range from our source population.

Experiment 3: To determine if adult quagga mussels were able to survive and grow in Truckee River water, the primary inflow into Pyramid Lake and the outflow for Lake Tahoe, we implemented another experiment with Lake Mead water as our positive control and Truckee River water (freshwater) as our treatment. Thirty adult mussels of similar size (15 ± 4 mm) were placed in individual containers with the treatment water being exchanged every 48 h to avoid the depletion of nutrients and food sources for the mussels. We checked physical parameters (temperature, salinity, conductivity, and dissolved oxygen) and mussel survival daily over the 58-day experiment. At the end of the experiment we weighed, measured, and extracted the gonads of all the mussels. We calculated the growth rate using the differences between the initial and final mass and length measurements divided by the number of weeks that the experiment lasted. To determine sex and reproductive viability, we dissected the mussel under a microscope to remove the gonad and placed the corresponding tissue into a histological cassette and preserved it in 10% formaldehyde solution. The samples were sent to the North Bay Histology Lab where permanent slides were produced. Each slide contained six tissue sections, 5 μm thick, stained with hematoxylin and eosin. We analyzed the slides using a compound light microscope (Meiji, Tokyo, Japan) with 100× magnification and a Lumenera Infinity II microscope camera to determine the sex and gonadal status. We differentiated three gonadal statuses according to the categories proposed by Guillou et al. (1990): (1) recovering (non-discriminable sex), (2) premature (discriminable sex), and (3) mature gonads (discriminable sex). We classified each mussel by gonad status, and when possible, by its sex. The percentage of reproductively viable individuals was determined following Davis et al. (2015) as:

% reproductively viable = (male count + female count)/(male count + female count + indeterminate count).

Survival of mussels exposed to Pyramid Lake brackish water with lower pH (short-term experiment)

Experiment 4: To test if the high pH of the Pyramid Lake water affected the survival of quagga mussels, we used water from Warrior Point (Pyramid Lake) with its natural pH values of 9.31 along with water titrated with phosphoric acid to reduce pH to the following concentrations 8.53, 8.83, 9.06, 9.21. Lake Mead water without pH alteration (8.6) was used as a control. Twenty-five adult mussels with a shell length range between 10–26 mm were selected for each water treatment, with all mussels from each treatment placed together in 4 L glass containers. When we observed a dead mussel, it was retrieved from the experiment and the time of death was recorded. The temperature during the experiment was reduced to 15 °C to decrease the metabolic rate and increase the mussels’ tolerance to higher salinity concentrations based on the observation made by Mills et al. (1996). We measured the pH using a handheld pH meter (Ohaus starter 300; Ohaus, Parsippany, NJ, USA) with a measurement resolution of 0.01 pH. We checked mussel survival hourly for 12 h, and after 24 h.

Survival of mussels exposed to Lake Mead and Truckee River freshwater with higher pH concentration (short-term experiment)

Experiment 5: To test if increasing pH reduced the survival of quagga mussels in water where they survive in natural conditions, we amended Lake Mead and Truckee River water with sodium hydroxide to increase the pH until reaching levels 9.0, 9.3, 9.5, 10, and 11. We used Lake Mead (pH 8.6) and Truckee River (pH 8.42) water without modifying the pH as control treatments. Thirty adult mussels with a shell length range between 12–24 mm were selected for each of the water treatments and mussels for each treatment were placed together in 4 L glass containers. The temperature in the chamber was set to 20 °C to assure high metabolic rates. We measured the pH three times a day using a handheld pH meter (Ohaus starter 300; Ohaus, Parsippany, NJ, USA) with a measurement resolution of 0.01 pH. We monitored the experiments continuously for 11 days.

Water quality monitoring and characterization

We routinely monitored the water quality during the experiments to determine the conditions under which quagga mussels can survive and grow. We measured temperature, pH, conductivity, salinity, and dissolved oxygen daily using a Y.S.I. digital handheld meter ProDSS. We analyzed the different treatments every other week for total hardness and alkalinity (phenolphthalein and total alkalinity) using a handheld titrator with EDTA 0.800 M and Sulfuric Acid Titration Cartridge 1.6 N, respectively (Hach Co, Loveland, CO, USA). Samples were filtered using glass fiber filters (Whatman GF/F filters with a pore size of 0.7 µm) and analyzed for dissolved anion and cations at the University of California, Davis Analytical Laboratory following standard operating procedures (https://anlab.ucdavis.edu/). Total nitrogen (TN) was analyzed at the University of California, Davis Analytical Laboratory by combustion following standard procedures with a detection limit of 0.1 mg L−1. Total phosphorus (TP) was analyzed at the High Sierra Water Lab following the EPA 365.3 method with a detection limit of 1 µg L−1. We determined chlorophyll a concentrations by filtering 150 ml of water through a Whatman GF/F filter and then measuring pheophytin-corrected chlorophyll a on a Turner 10-AU fluorometer following methanol extraction (Marker, 1972).

Statistical analysis

The data were expressed as means ±1 standard error. All statistical analyses were completed using R software (R Core Team, 2020) and graphics were plotted using the ggplot package (Wickham, 2016). We used linear regression to analyze mortality and growth rates in relation to mussel size. We quantified differences in growth (length and weight) between Truckee River and Lake Mead treatments using linear mixed-effects models (LMM), with site modeled as a fixed effect. To assess differences in sexual maturity between Truckee and Mead water treatments, we compared the percentage of each sex and gonad status for each water treatment. Differences in mortality rates between Truckee River and Lake Mead mussels under different pH treatments were tested using a LMM with site and pH modeled as fixed effects. All models met the assumption of homoscedasticity. For all the models, we considered a significance level (alpha) of 0.05. Tests for significance of the fixed effects and their interactions in the models were performed via the Wald statistic (Zuur et al., 2009) using the gls and lme functions in the nlme R package (Pinheiro et al., 2020). Multiple comparisons among sites and pH were performed with Tukey’s HSD post hoc test (emmeans R package; (Lenth, 2021)).

Results

Water quality monitoring and characterization

The two Pyramid Lake locations (Popcorn Rock and Warrior Point) were similar with higher salinity, pH, total alkalinity and hardness compared to the Truckee River and Lake Mead (Table 1). The quality of the water in the storage tanks for each of the lakes and treatments remained stable throughout the duration of the experiments (Table 1). The highest chlorophyll a concentrations, that were used as an estimate of algal biomass, were found at Warrior Point (0.9 ± 1.3 µg L−1) and Popcorn Rock (0.8 ± 1.1 µg L−1). However, for all treatments, we observed a stark increase in algal growth when water was transferred from the tanks to the experimental chamber with increased light and regulated temperature. There was slightly higher salinity at Warrior Point (4.3 ± 0.1ppt) compared to Popcorn Rock (4.1 ± 0.1 ppt), which is closer to the Truckee River inflow (Table 1), and the pH values were 9.3 at both sites (Table 1). Truckee River water had lower pH (8.1) and salinity (0.2 ± 0.0) and Lake Mead water had lower pH (8.3) and ten times lower salinity (0.4 ± 0.0 ppt) compared to Pyramid Lake. Additional parameters such as total alkalinity, hardness, dissolved oxygen, calcium, and others can be found in Table 1.

Survival, growth, and reproduction of mussels in Pyramid Lake brackish water

In Experiment 1, with Warrior point and Popcorn Rock treatments, all but one of the adult mussels died after 19 h, whereas all mussels survived in the positive control treatment (Lake Mead water). Similar results were found in the Experiment 2 when analyzing the effect of mussel size on survival. Approximately 90% of the medium-sized mussels died after 4 h in Popcorn Rock and Warrior Point water, and approximately 90% of all mussels were dead within 9 h (Fig. 2). After 12 h, almost all the mussels in the three different size classes died in Warrior point and Popcorn Rock treatments, with a 90% mortality in large sized mussels after 12 h. Only one mussel of medium size survived for more than 24 h in the Popcorn Rock water. Conversely, in Lake Mead water treatment (Experiment 2), we recorded only one mussel death after 12 h. The relationship between size (weight and length) and time showed high dispersion with mussels of different sizes dying indistinctly every hour. The body tissue of the dead mussels retrieved after 19 h had a bloated appearance with some exhibiting tissue disintegration (Fig. 1F). The mussels’ shell gape closure time in response to touch was very slow before death.

Figure 2 Cumulative frequency of survival in the Lake Mead, Popcorn Rock, and Warrior Point treatments in three different class sizes: large (16–26 mm), medium (6–16 mm), and small (3–6 mm).

Survival, growth and reproduction of quagga mussels in Truckee River fresh water

Quagga mussels were able to survive, grow, and reach reproductive maturity in the freshwater of the Truckee River. All 30 mussels survived in the Truckee River water after 58 days of the experiment, while two individuals died in the Lake Mead treatment (positive control). The increase in weight (mean difference: 0.02) and length (mean difference: 0.27) of the mussels was double in the Truckee River treatment compared to the Lake Mead treatment (mean differences: 0.01 and 0.11 for weight and length, respectively) (Figs. 3A and 3B). The mean growth rate of adult mussels was 0.0402 g/week (p < 0.001, R2 = 0.53, linear regression) for Truckee River water treatment (Fig. 3B), and −0.0026 g/week (p = 0.94, R2 < 0.01, linear regression) for Lake Mead treatment (Fig. 3A). Most of the mussels increased their size and weight in the Truckee River treatment. There was a positive and significant correlation between the differences in mussel weight and length between the start and the end of the experiments (Truckee: p < 0.001, R2 = 0.53; Mead: p = 0.94, R2 < 0.01, lineal regression; Figs. 3A and 3B). The mussels in the Lake Mead treatment showed only slight changes in length and weight between the start and end of the experiment, and there was no correlation between the increase in these variables (p = 0.94, R2 < 0.01, linear regression; Fig. 3A). The differences in length and weight between the Mead and Truckee water treatments were not significant (p = 0.12 and p = 0.11, respectively, linear mixed-effects models).

Figure 3 Linear regression and barplot for analysing growth and reproductive potential.

Linear regression between the difference in length and weight between the start and the end of the experiments in Lake Mead with numbers indicating individual mussels in the experiment (A) and Truckee River (B) treatments. Barplot showing the sex and gonads status from adult mussels after 58 days of being in Lake Mead (C) and Truckee River (D) water. n.d: no determined sex.

Surviving adults showed high reproductive viability at the two sites (Figs. 3C and 3D), but we observed higher reproductive viability in the Truckee River treatment (93.3%) compared to the Lake Mead (78.5%) treatment (Fig. 3C). In the Truckee River treatment, 80% of the adult mussels were in a reproductively mature state, ~13% in a premature state, and ~7% were immature or in a recovering state (Fig. 3D). In the Lake Mead treatment, ~32% of the adult mussels were in a reproductively mature state, ~46% in a premature state, and ~21% were immature or in a recovering state (Fig. 3C).

Survival of quagga mussels exposed to Pyramid Lake brackish water with lower pH

Lowering the pH of Pyramid Lake water to values similar to Lake Mead, where they have established a permanent population, did not result in higher mussel survival. Between 90–100% of the mussels exposed to different pH concentrations at Warrior Point water treatment were dead after 24 h (Fig. 4), while all mussels survived in the control treatment. The mussels’ survival rate at pH levels below 9.0 was lower than at pH levels higher than 9.0. Two of the twenty-five mussels survived in each of the experiments with pH 8.53 and 8.83 after 24 h, one of twenty-five mussels survived at pH 9.31 after 24 h, and all the mussels died at pH 9.21 and 9.06 after 24 h. In all of the treatments more than 60% of the mussels died after 10 h, except for pH 8.53, where 40% of the mussels died during that period. During the experiment, the mussels in Warrior Point treatment appeared more stressed (partially opened valves and did not extend the siphon for filtering) compared to those in Lake Mead control water treatment.

Figure 4 Cumulative frequency of survival in Warrior Point (Pyramid Lake) water treatment amended with phosphoric acid to obtain different pH concentrations.

Lake Mead water treatment was the control treatment.

Survival of mussels exposed to Lake Mead and Truckee River freshwater with higher pH concentration

Increasing the pH of freshwater resulted in higher mussel mortality rates at very high pH (pH > 10), however, the mortality rates at any pH in freshwater were very low compared to those found in the brackish Pyramid Lake water. All mussels were dead after 6 days (Truckee River) or 9 days (Lake Mead) in the treatments where we increased the pH to 11 (Fig. 5). After 10 days, only 15% of the mussels died at pH 10 in Mead and Truckee water treatments, but after 14 days, all the mussels died at pH 10 (Fig. 5). We also observed no significant differences between Lake Mead and Truckee River at pH 10 and 11 (p = 0.50, linear mixed-effects models), but there were significant differences between the pH 10 and 11 (p < 2e−16, linear mixed-effects models) in each of the water treatments. We also noted low mortality in other pH levels. Only one mussel died after 14 days at pH 9.0 in Lake Mead and at pH 9.0 and 9.3 in Truckee River water (Fig. 5). No mortality was recorded in the controls, pH 9.5, and pH 9.3 in Lake Mead water (Fig. 5). In Lake Mead water treatments of pH 9.5, 10, and 11, we observed carbonate precipitation, but this was not observed in Truckee River water at any pH. The mussels had no visual signals of stress in either Mead or Truckee water. In our daily observations, we observed that the mussels exhibited less stress in these treatments with freshwater compared to the experiments where we used the brackish water of Pyramid Lake.

Figure 5 Cumulative frequency of survival in Lake Mead and Truckee River freshwater treatments amended with sodium hydroxide to obtain higher pH concentrations.

Lake Mead and Truckee River water treatment without amendments were used as a control.

Discussion

Adult quagga mussels can survive, grow, and reach sexual maturity in the freshwater of the Truckee River, but not in the brackish waters of Pyramid Lake (Fig. 6). A reduction in the pH of the Pyramid Lake brackish water did not improve survival of the mussels (Fig. 6). Similarly, increasing the pH of the Truckee River water did not result in higher mortality rates (Figs. 4 and 6). These results suggest that the chemical composition of Pyramid Lake’s water (particularly salinity and ion concentrations) could be the primary reason that quagga mussels cannot survive in this lake with the relationships between growth and shell length are indicative of stressful conditions for mussels (Claudi et al., 2013). In our experiments, we observed increased mussel growth with no evidence of stressed conditions in Truckee River water compared to Lake Mead water. Furthermore, our results align with Davis et al. (2015) who found that adult quagga mussels can settle, grow, and survive in the upper basin of the Truckee River. These combined results indicate that adult quagga mussels may have a chance of establishing in the lower Truckee River if they were to be introduced and given optimal conditions for growth. We did not assess the influence of physical turbulence to the maintenance of mussels which could be important during the winter and spring snow melt conditions of the region. In addition, our study analyzed the reproductive potential and not the capacity of quagga mussel veligers to complete the life cycle in the Lower Truckee River, which would be needed to establish a permanent population. Thus, the invasion potential could be restricted by other environmental variables that prevent quagga mussel veliger survival and settlement (such as flow velocity, water temperature, and substrate composition) that were not analyzed in this study (Chen et al., 2011).

Figure 6 Conceptual model with the main results from this work and rising concern.

The Truckee River presents a high risk for quagga mussel infestation by adults if the veligers are able to survive in the calcium (Ca), alkalinity, and total hardness conditions of the water. Results from Pyramid Lake water’s Ca, alkalinity, and total hardness values were not conclusive enough to use as exclusive predicters of potential quagga mussel invasion. Calcium tolerance for quagga mussels has been based on presence/absence and abundance distribution data, suggesting that quagga mussels are less tolerant to lower Ca concentrations than zebra mussels, but the tolerances of both species mostly overlap (Jones & Ricciardi, 2005; Mackie & Claudi, 2010; Karatayev & Burlakova, 2022). The concentration of Ca in Pyramid Lake is low (8.2–8.8 mg L−1), indicating a low risk of invasion given the limits proposed by Mackie & Claudi (2010). However, one experimental-based analysis showed that quagga mussels have the potential to establish in Lake Tahoe, which has low concentrations of Ca at 9 mg L−1 (Davis et al., 2015). In addition, sink populations of quagga mussels on the St. Lawrence River below the Ottawa River confluence have been favored by its hydrological connectivity with the Great Lakes that have higher Ca concentrations (Cohen, 2007). Alkalinity and hardness values in Pyramid Lake are also higher than the limits proposed by Mackie & Claudi (2010) for systems considered at high risk of invasion, but the optimal ranges in their study were based on zebra mussels. Although pH, Ca, hardness, and alkalinity are correlated, the relationship is not strong, and Ca can originate from other compounds that do not contribute to total alkalinity (Mackie & Claudi, 2010). Thus, limits of hardness and alkalinity tolerances could differ slightly from those found for zebra mussels since calcium and pH limiting values have been poorly studied for quagga mussels specifically.

Salinity concentrations in Pyramid Lake are in the upper ranges of tolerance observed for quagga mussels (Mackie & Claudi, 2010; Garton, McMahon & Stoeckmann, 2014). Therefore, we suggest that the current global method for estimating the potential invasion of quagga mussels based on chalk variables (Ca, pH, Total alkalinity, and Total hardness) should also include additional variables such as salinity and Na and K ion concentrations, especially in the case of fresh-brackish waters or terminal lakes. Having specific case studies of survival across mussel life history will be valuable for waters with distinctly different ionic characteristics. Quagga mussels of all sizes died after 12 h of exposure to brackish Pyramid Lake water, even when we reduced the temperature to 15 °C to decrease the metabolic rate. Such a rapid mortality rate has not been previously reported. The swelling and disintegration of body tissues suggest salinity and pH as probable causes of mortality (Claudi et al., 2013). The osmoregulation system of quagga mussels is adversely affected by high salinity and pH (Claudi et al., 2013). The salinity (4–4.3 ppt) and pH (9.31) in both Pyramid Lake treatments could be sufficiently elevated to cause high mortality in mussels (Aleksenko, 1991; Spidle, Mills & May, 1995; McMahon, 1996). Other experiments with mussels in North America indicate that the maximum salinity tolerance is in between 4–5 ppt with a mean survival time of 9–14 days at 15 °C and 5 days at 20 °C in waters with salinities of 5 ppt without acclimation (mussels transferred from 0 to 5 ppt) (Aleksenko, 1991; Spidle, Mills & May, 1995; McMahon, 1996; Hofius, Mandella & Rackl, 2015). Acclimatization to a prior salinity of 2 ppt could increase tolerance to higher salinities. However, both acclimatization and non-acclimatization experiments showed values of salinity above 4 ppt are lethal for mussels (Kilgour et al., 1994). The high rate of mortality we observed in our experiments compared to previous studies suggests that factors other than salinity may affect survival.

The high pH of Pyramid Lake water was not the cause of high mortality of quagga mussels. Claudi et al. (2013) observed tissue damage in adult quagga mussels exposed to high pH concentration and showed higher sensitivity to pH relative to zebra mussels (Claudi et al., 2013). Quagga mussels presented 100% mortality in pH 12 in less than 15 h (Claudi et al., 2013). However, in the experiment where the pH of brackish water was decreased to 8.4, we still observed 100% mortality after 24 h. Like Claudi et al. (2013), we did not observe mortality after 14 days of exposure to freshwater at pH ≤ 9.1, 9.3, and 9.5, but there was a 50% mortality at pH 11 after 24 h in fresh water (Truckee River) where mussels were able to survive for 6 days. Therefore, we concluded that quagga mussels are sensitive to high pH in fresh waters, but high mortality of this species in brackish water might be attributed to other chemical and ionic characteristics.

Dreissena species are sensitive to ionic concentrations such as chloride, sodium, and potassium. The survival of zebra mussels in higher salinities depends on the balance of K and Na that allows them to eliminate water gained over an elevated osmotic gradient (Dietz & Byrne, 1997). Low concentration of K can kill zebra mussels within 6–24 h by inhibiting respiration and reducing filtration rates (Fisher et al., 1991; Horohov et al., 1992). Zebra mussel veligers cannot settle, adults cannot attach to substrates at concentrations of K as low as 10–30 mg L−1 (Fisher et al., 1991), and their valve closure is inhibited by high concentrations of K around 70–80 mg L−1 (Wildridge et al., 1998). The concentration of K at Pyramid Lake is about 126 mg L−1 (Reddy & Hoch, 2012), so the slow reaction to touch exhibited by quagga mussels in our experiments when monitoring for survival could be the result of stress induced by high concentrations of K. In a study by Horohov et al. (1992), zebra mussels died after one week of exposure to low concentrations of NaCl (40 meq L−1). At Pyramid Lake, the concentration of NaCl is ~50 meq L−1. The survival of a few individuals for more extended periods in the different experiments can be explained by the mussels’ ability to sense and respond to toxins by closing their shell valves for periods greater than 48 h (Rusznak, Mincar & Smolik, 1994).

Salinity and specific ion (K, Na, and Cl) concentrations may explain the survival of quagga mussels in Truckee River water, and their high mortality in Pyramid Lake water. The historical record of evolution and dispersal of dreissenid mussels provide evidence of their high potential to adapt to extreme environments; for example, Mills et al. (1996) showed that after a few generations these mussels evolved to be more tolerant to salinity in Europe. Given the so-called rapid evolution, or perhaps genetic drift, of quagga mussels a question still remains: if Dreissena spp. establish in the Truckee River with hydrologic connections to the Pyramid Lake ecosystem, can there be adaption and colonization of quagga or zebra mussels to new habitats outside of their existing distribution? Yokomizo & Takahashi (2020) found that in systems with different environmental conditions next to each other at relatively small spatial scales, individuals that can adapt to the environment could expand and colonize new habitats. A clear example of this potential situation is the case of the Colorado River system, where indirectly connected sites favored unprecedented high rates of quagga mussel spread in the western U.S. This connectivity facilitated establishment of quagga mussel populations under variable environmental conditions (Wong & Gerstenberger, 2011; Pucherelli et al., 2016). Invasive species, such as the Mytilopsis spp (recognized as the brackish water equivalent of zebra mussels), have shown differences in their biological traits, environmental conditions, and substrate utilization between their native and non-native habitats, which has been attributed to their wide plasticity and ability to adapt to new conditions in the invaded system (Rodrigues et al., 2022). As alternatives to species plasticity or broad physiological tolerances, other studies have suggested that invasive species are genetically dynamic populations over short periods of space and time (Lee, 2002). After invasions, genetic drift and natural selection to local environmental gradients could alter the genetic structure of the invading populations, which could favor establishment success through modifications to their tolerances and/or behaviors, resulting in rapid evolutionary events (Lee, 2002). Thus, it well could be that connected ecosystems with contrasting environmental characteristics could favor genetic adaptation that leads to a shift into novel niches in the invaded ecosystem. Future studies should consider exploring genomic characteristics of invasive species in addition to ecological preferences in native and non-native species.

Most of the studies analyzing the ranges of water quality parameters that favor the invasion of dreissenid mussels are based on the tolerances of zebra mussels (Mackie & Claudi, 2010; Karatayev & Burlakova, 2022). Furthermore, those studies found different limits of tolerances of Dreissenid mussels for parameters such as salinity, temperature, substrate, and dissolved oxygen (Mackie & Claudi, 2010). However, the environmental conditions such as pH, Ca concentration and salinity in locations where quagga mussels can establish a permanent population are still scarce (Karatayev & Burlakova, 2022). Studies of zebra and quagga mussel tolerances for salinity were made in the eastern region of the U.S. in the 1990s, and after 30 years of invasion, a reassessment of the environmental conditions, in particular salinity, is needed as we should not dismiss the possibility of an invasion of quagga mussels in waterbodies with higher salinity than previously described. Therefore, we recommend that additional studies assess the environmental conditions that favor quagga mussels’ invasion and their tolerance to physical and chemical conditions. Perhaps one or a few parameters may not be enough to determine the potential for quagga mussel invasions and specific site studies are needed.

Conclusions

Adult quagga mussels can survive, grow, and reproduce in the Truckee River’s freshwater conditions, but not in the brackish waters of Pyramid Lake. In Pyramid Lake, decreases in pH and temperature did not reduce their mortality rate and we observed mussel mortality in all size ranges. Our results suggest the chemical condition (especially salinity and ion concentrations) of Pyramid Lake water may be the main reason quagga mussels cannot survive in this lake. Mussels exposed to Truckee River water exhibited positive growth and presented no visual evidence of stress. Mortality was not observed in increased pH of the fresh, Truckee River water and Lake Mead water, except at pH > 10. Adult quagga mussels can tolerate pH as high as 10 for 14 days, while pH 11 caused 100% mortality after 6 or 9 days, and no mortality was observed at pH < 9.5 during the 14 day experiment. The results of the high pH experiment from this work are an important contribution to fill the gaps in knowledge about the upper tolerance limits of adult quagga mussels.

Hydrological connectivity between fresh and brackish inland ecosystems can favor the adaptation and colonization of quagga or zebra mussels to new habitats outside of their existing ranges. After 30 years of invasion and evolution, the quagga mussels’ environmental tolerances are still uncertain; this fact, contrasted with the possibility of quagga mussel invasion of higher salinity waterbodies, suggests the need for further investigations of the impact of water physical and chemical conditions on their invasion potential, especially regarding salinity. Also, our results suggest that one or a few parameters may not be enough to determine the potential for quagga mussel invasion, and specific site studies are needed.

The authors would like to thank staff from the Pyramid Lake Paiute Tribe were invaluable in their knowledge of the lake and assistance in the field: Dan Mosley, Denise Shaw, Robert Eagle, Jon Cawelti, Jennessy Toribio, Adrienne Juby, Justin Jackson, and Sloan Sampson. We also want to express our appreciation to personnel from the Nevada Department of Wildlife, Kevin Netcher, Brandon Senger and Amos Rehm, and from Southern Nevada Water Authority Todd Tietjen, Deena Hannoun, and Roslyn Flanagan for providing access to Lake Mead water and quagga mussels. We thank three anonymous reviewers for comments that improved this manuscript.

Additional Information and Declarations

Competing Interests

Author Contributions

Field Study Permissions

Data Availability

The authors declare that they have no competing interests.

Carina Seitz conceived and designed the experiments, performed the experiments, analyzed the data, prepared figures and/or tables, authored or reviewed drafts of the article, and approved the final draft.

Facundo Scordo conceived and designed the experiments, performed the experiments, analyzed the data, authored or reviewed drafts of the article, and approved the final draft.

Erin Suenaga performed the experiments, authored or reviewed drafts of the article, and approved the final draft.

Emily M. Carlson performed the experiments, authored or reviewed drafts of the article, and approved the final draft.

Shaye McMillen performed the experiments, authored or reviewed drafts of the article, and approved the final draft.

Logan Gregory performed the experiments, authored or reviewed drafts of the article, and approved the final draft.

Sudeep Chandra conceived and designed the experiments, performed the experiments, analyzed the data, authored or reviewed drafts of the article, and approved the final draft.

The following information was supplied relating to field study approvals (i.e., approving body and any reference numbers):

Collection of specimens and water was approved by the Nevada Department of Wildlife

The following information was supplied regarding data availability:

The data is available at Mendeley Data: Seitz, Carina; Scordo, Facundo; Suenaga, Erin; Carlson, Emily Margaret; McMillen, Shaye; Gregory, Logan; Chandra, Sudeep Chandra (2022), “Water salinity and pH effect on invasive quagga mussel survival, growth, and reproduction”, Mendeley Data, V1, DOI 10.17632/p9ssvv59mx.1.

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
