# Peer review of "Salinity and pH effects on survival, growth, and reproduction of quagga mussels"

_PeerJ, doi:10.7717/peerj.15450_

## Round 0.1 · original submission · Major Revisions

Three recognized experts have assessed your manuscript and identified a number of issues that make the manuscript unacceptable in its present form. On the other hand, all three reviewers praise the quality of your experimental work.

An important aspect to be considered in your revision is a thorough improvement of the English usage and grammar. In its present version, the manuscript is very difficult to read. The reviewers provide suggestions to improve the writing (including annotated manuscript file by reviewer 3) but these suggestions are exemplary rather than comprehensive. Further issues are that results should be interpreted in the discussion chapter more cautiously (see reviewer 1), that a more thorough and updated review of the literature is needed in the chapters introduction and discussion and more details regarding material and methods are required (reviewer 2).

I hope that the detailed evaluations by the three reviewers will allow you to carry out a substantial revision of your manuscript, which is a precondition for its acceptance.

Reviewer 1 ·

Basic reporting

In the Introduction the authors mention the discovery of quagga mussels in the western US, including in Colorado and Utah. It should be noted that all populations discovered in these two states have failed to persist (see https://nas.er.usgs.gov/viewer/omap.aspx?SpeciesID=95), although a population of zebra mussels has recently been discovered in Highline Lake in Colorado.

Language: The language in the manuscript should be improved to ensure clarity. There are numerous grammatical issues throughout the manuscript, particularly with regards to singular/plural nouns and versions, and definite/indefinite articles. A list of edits is provided below, but is by no means exhaustive. I would suggest that the manuscript be carefully edited for clarity of language by a colleague who is proficient in English and familiar with the subject matter.

Minor edits (this list is not exhaustive):

Genus and species names should be italicized throughout.

Line 81: "sinks" should be changed to "sink"

Line 83: "quagga mussels' veliger" should be changed to "quagga mussel veligers"

Line 101: "tolerances from" should be changed to "tolerances is from"

Line 103: I assume "the terminal basin" refers to Lake Tahoe Basin? This should be stated explicitly for clarity.

Line 103: "Are" should be changed to "is"

Line 104: "a" should be removed from "a site specific studies"

Line 121" "water of" should be changed to "water from"

Line 165 "(details below." should be changed to "(details below)."

Line 168: "experiment" should be changed to "experiments"

Line 227 AND 237: should "controlled" be "measured"?

Line 246: "Linear regression" should be changed to "The linear regression"

Line 328: This sentence should be reworded. I assume "their natural conditions" refers to conditions where quagga mussels are established (e.g. Lake Mead)? This should be clarified.

Line 331: I would suggest changing "was higher" to "has greater" so that "higher" does not appear twice in the sentence in different contexts.

Line 350: "treatment" should be "treatments"

Line 374: "combine" should be changed to "combined"

Experimental design

The experiments presented were well designed, with measurement and reporting for relevant variables in water chemistry.

Methods were reported with sufficient detail for replication.

Validity of the findings

Relevant data are reported for the experiments and statistical analyses employed are appropriate.

The discussion includes a range of variable that may account for differences in mortality observed across experiments. However, the authors may overstate their case when they say on lines 399-402 that "we consider the current global method of estimating the potential invasion of quagga mussels
based on chalk variables (Ca, pH, Total alkalinity and Total hardness) not appropriate for water
since there is a knowledge gap regarding the survival of this species." The data presented in the current paper do not seem to be at odds with chalk variables presented by Mackie & Claudie, 2010. In fact, the alkalinity reported for Pyramid Lake as the upper limit by Mackie & Claudie. While low alkalinity is generally considered a limiting factor, the high alkalinity observed could also be linked to physiological stress.

It might be better stated that these variable should not be taken in isolation without measuring and reviewing other factors that have been shown to contribute to quagga mussel survival. As discussed in the manuscript, ions such as Na and K were were at levels in Pyramid Lake that have been shown to be toxic to quagga mussels, and these likely contributed to the rapid mortality observed. It would therefore seem that it is important to measure these variables and include them in considerations of habitat suitability. Although it should be noted that these elevated values may be related to the fact that Pyramid Lake is a terminal lake, and such high ion concentrations will most likely be observed in comparable waters.

Reviewer 2 ·

Basic reporting

The entire manuscript would benefit from thorough copy editing as there are several instances of missing words, grammatical errors, and syntax that could be improved for readability and clarity. A non-exhaustive list of examples includes line 89 has awkward syntax (“as regards which”), lines 101-105 is a run-on sentence, the methods section contains multiple instances of randomly capitalized words (e.g. line 234 - Sodium Hydroxide), and the results contain conflicting and confusing information (lines 331-336 describe conflicting mortality rates). Also be sure that scientific notation is correct - line 146 contains a “u” in place of a mu symbol, and ensure that chlorophyll a has a consistently italicized “a” (e.g. line 151).

The introduction and discussion could benefit from a more thorough and updated review of the literature on this topic. For example, on line 55-56 a 2005 source predicts $1 billion annually in damages from dreissenids (location is unspecified). It is 2022, 17 years later - is there an update on this number? Have dreissenids cost $1 billion annually somewhere? Lines 60-71 address the invasion timeline and intensity of both dreissenid species but does not mention that quagga mussels have been outcompeting established zebra mussel populations in the Great Lakes basin for over a decade (see Ram et al. 2011 https://doi.org/10.1080/07924259.2011.588015). Starting on line 76 the authors report that pH and Ca metrics have been rarely evaluated on quagga mussels, however a cursory literature search found several relevant articles that are not mentioned in the current manuscript (see Baldwin et al. 2012 https://link.springer.com/article/10.1007/s10530-011-0146-0; Whittier et al. 2008 https://doi.org/10.1890/070073; Gopalakrishnan and Kashian 2019 https://doi.org/10.1002/etc.4624). Finally, for discussion on the evolutionary relevance of ionic stress in brackish-freshwater invasions, I suggest looking into work by Carol Eunmi Lee from University of Wisconsin-Madison and a recently published paper by Rodrigues et al. (2022 https://doi.org/10.1007/s10530-022-02772-z).

Line 84 is a statement that needs a citation.

Experimental design

In general, the methods would benefit from simplification and clarification. Line 158 - how were the mussels removed from the water and at what depth? Lines 163-164 have a verb tense disagreement with the rest of the methods and the authors also need to describe what size (length and/or weight) of mussels were used in experiments, or the authors could remove this redundant information in this section and provide sufficient description in the experimental design sections. Line 164 - what conditions would cause a mussel to be placed in either a 13 or 4 L container? This is a big size discrepancy so need to describe why the authors chose two different size containers for different experiments. Is it correct that water was stored in 2273 L tanks? That’s huge! Line 173 - mussels were dried then discarded? Did you weigh them after they were dried? Also, what did you dry? The entire mussel or just soft tissue?

In the experimental design, the authors should elaborate as to why different numbers and lengths were selected for different replications and why experiments were conducted for varying times (e.g. lines 188-196). It should be clear how many individuals of what size range were placed in which treatments for what length of time, and what variables were measured (e.g. line 223-224 it is unclear if 25 total mussels were distributed among 6 treatments of if 25 mussels were in each treatment, and if mussels were pooled or individually tested). Line 185 describes Experiment 1 and Line 195 describes Experiment 3, but there is no Experiment 2. Why were the pH experiments in Pyramid Lake Water performed at different pH than the Lake Mead and Truckee River experiments? In the Pyramid Lake experiment, the Lake Mead treatment would not be a true control because it does not have the same salinity of the brackish waters. Why were the two pH experiments (freshwater vs brackish) performed at two different temperatures?

Lines 145-146, 150: Whatman GF/F filters have a pore size of ~ 0.7 microns, not 47 microns. The 47 on the filter container refers to the diameter in mm of the filter.

The statistical analyses need more detail - what was your alpha value, what models did not meet assumption of homoscedasticity? Salinity and reproduction are not mentioned in this section despite being major variables and outcomes of the experiments. How were the effects of salinity analyzed? How were differences in reproductive maturity assessed among different treatments? The pH experiments need to include analysis of salinity as well as it could be a compounding factor in mortality.

Validity of the findings

The results would be stronger if they were more clear and concise. I suggest removing redundant information from the text of the results that can be found in a table - for example, lines 270-277 is information that can be found in Table 1 and does not need to be re-written. All subsections of the results need to include results of statistical analysis (p-value, analysis run, any other important statistics) and numerical results instead of comparative words (Lake Mead had slight changes in weight, with an average increase of X g/week (p=0.1, R=0.9, GLMM)). Watch for conflicting statements, for example Line 293 is in conflict with line 287. Paragraph 328-338 is also confusing and has seemingly contradictory information. Lines 266-268 need clarification - this is the first indication that water storage tanks were opaque (this belongs in methods). Please provide chl-a values to validate this statement. Were the chl-a increases at a high enough concentration and variable within the treatments to warrant including this value in the analysis? What kind of algal growth was it, as a toxic cyanobacteria may compound poor health effects from your other treatments. In the paragraph starting on line 286, please provide more detailed information about which mussels died in which treatments (e.g. Warrior Point had 90% mortality after X hours while Popcorn Rock had 100% mortality after Y hours). This paragraph also needs statistical analysis information on the data. The paragraphs starting at lines 303 and 317 also need statistical analysis information, reference to the table or figure in the manuscript, and numeric values to accompany the comparisons between treatments. For lines 307-311, how did you determine that this variance in weight wasn’t due to water on the mussel? It is difficult to accurately weigh live mussels because of the water weight, and this method was not described in the methods section (i.e. did you dry off the outer shells prior to weighing?). Lines 354-356: was this carbonate precipitation measured? Please explain how this could affect the mussels or water chemistry in the treatments. Lines 356-358 describe examining mussels for stress and also a daily survey; neither of these are explained in the methods so please do so. Figure 3A&B needs a better descriptor - the y-axis should be “change in weight” according to the manuscript. The legend for Figure 5 has 7 variables, but only 4 are shown on the plot.

The authors need to be careful about how they represent their results in the discussion. Line 363 states that the results demonstrate quagga mussels can reproduce in freshwater, but the authors did not evaluate reproduction via spawning or fertilization - the authors evaluated gonad maturity, or reproductive potential. Lines 374-376 discuss potential for quagga mussel population establishment, however the experiments in this manuscript did not evaluate that. Mortality of individuals is not the same as population establishment potential. On lines 447-449, the authors state their experiments suggest quagga mussels survival depends on specific ions, but specific ionic concentrations were not evaluated in this study. Further, the authors list K as an ion that determines quagga mussel survival, but potassium was not evaluated in any of the water samples according to Table 1. Lines 465-468 are untrue; there is plenty of literature that describes the environmental range of quagga mussels and the conditions under which populations have established. There is even a profunda morphology of quagga mussels that have established up to 130 m underwater. There has been a lot of research on dreissenid tolerance (to various chemicals, environments, etc.) since the 1990s outside of the Eastern US. Please revise this and be more specific about which conditions you are referring to. I would also like to see a bit more inference made from the results. For example, in lines 393-394 please elaborate on what this incongruity in alkalinity and hardness values means for your study and quagga mussel invasion in general. Lines 406-407 also need elaboration and a source for this information. Lines 429-430 also need elaboration on what ions or chemicals you are referring to and this statement would be boosted by some statistical analysis of how salinity works in tandem with pH on mussel survival.

Additional comments

I appreciate the work put into this research project. It provides important information on the habitat range of invasive quagga mussels in the western United States and may be useful for management. Overall, this manuscript would benefit from a more thorough and updated literature review, clarification and detail, and copy editing.

Reviewer 3 ·

Basic reporting

The manuscript describes the results of a very interesting laboratory study of the ability of quagga mussels (Dreissena rostriformis bugensis) collected from infested Lake Mead to survive and grow in fresh water from the uninfested Truckee River and in brackish water from two sites in Pyramid Lake in Nevada. The results indicated that quagga mussels could survive, grow and reproduce when maintained in freshwater from Lake Mead and the Truckee River but not in saline water (4 ppt) from Pyramid Lake. The Pyramid Lake sites had a higher pH (9.3) than Truckee Lake (8.1) and Lake Mead (8.3) but even when the Pyramid lake water ph was lowered to that of Lake Mead and the Truckee River, the large majority of mussels in Pyramid Lake water (<10%) did not survive. This result indicated that factors other than pH alone were inducing mortality in quagga mussels held in Pyramid Lake water. In contrast, 100% mortality occurred in Lake Mead or Truckee River mussels when exposed to pH artificially increased to 10 and 11 while little or no mortality occurred in mussels exposed to pH’s of 9.5, 9.3, and 9.5.

The results indicated that quagga mussels were likely to be able to establish a sustainable population if introduced to the Truckee River while the saline conditions in Pyramid Lake were likely to make it highly resistant to establishment of a sustainable quagga mussel infestation. Such information is important when developing risk assessments for quagga mussel invasion of southwestern water bodies. In addition, almost all testing of environmental tolerances has been done on zebra mussels (Dreissena polymorpha) with far fewer environmental tolerance studies available for quagga mussels. Available quagga mussel studies have indicated that there are differences in the environmental tolerances of the two species suggesting that further studies such as described in this manuscript are required to accurately develop quagga mussel invasion risk assessments in North American water bodies. As such, the information presented in the manuscript should be of interest to readers of PeerJ, especially those working in area of aquatic invasive species and particularly those involved with the study, dispersal, and prevention of Dreissenid mussels in North America.

Experimental design

The experimental design, analysis of results, discussion and citations appeared appropriate. Statistical approaches to data analysis appeared sound. Captions for Figures 1-6 and the legend for Table 1 were not provided with the materials sent to this reviewer. However, the figures and table appeared to be appropriate for the information presented and easy to understand.

Specific Comments:

1. Page 1, lines1-2: The title seems a bit clunky. Suggest changing it to “Salinity and pH Effects on Survival, Growth, and Reproduction of Quagga Mussels.”

2. Page 6, line 222: The shell length range of mussels used in the pH tolerance experiment should be provided here.

3. Page 8, line 293: There is no citation of Figure 2 in the text. The citation to Figure 3 in this line appears to actually that for Figure 2.

4. Species names must be italicized throughout the References section.

5. Table 1: Change “Dissolve Oxygen” to “Dissolved Oxygen”.

Validity of the findings

Findings appear to be valid and well supported.

Additional comments

While the research described in the manuscript is interesting and important, the manuscript is in need of extensive editing as there are many grammatical and English usage problems throughout. This reviewer has made an extensive effort to help improve the grammar and readability of text by providing suggestions for editorial corrections and improvements in the “Minor Editorial Comments and Suggestions to Improve Grammar and Readability” section of this review below. However, it is recommended that any revision include careful editing to improve grammar and sentence structure.

All comments below refer to specific pages and line numbers in the manuscript. The manuscript with line numbers has been attached to this review to allow easy access to portions of the manuscript for which comments for change or improvement have been provided.

Minor Editorial Comments and Suggestions to Improve Grammar and Readability

1. Page 1, line 22: Change “Great Lake” to “Great Lakes”.

2. Page 1, line 23: Change “western water” to ‘western water bodies”.

3. Page 2, line 39: Change to “---environmental tolerances appears to be essential---“

4. Page 2, lines 49-50: Change to “---invasive zebra (Dreissena polymorpha (Pallas)) and quagga (Dreissena rostriformis bugensis---“ properly abbreviating species names.

5. Page 2, line 61: Change “Zebra mussel was discovered” to “Zebra mussels were discovered” to improve grammar.

6. Page 2 line 62: Change “Quagga mussel were found” to “Quagga mussels were found” to improve grammar.

7 Page 3, line 80: Change to “showed that quagga mussels” to improve grammar.

8. Page 3, line 81: It is not clear what “permanent sinks populations is referring to in this sentence. It would be better to remove sinks and write “permanent populations of adult quagga mussels”.

9. Page 3, line 84: Change “the potential quagga mussel's invasion” to “the potential for quagga mussel invasion”.

10. Page 3, line 89: Change to “results regarding the upper pH limit tolerated by quagga mussels tolerate” to improve readability.

11. Page 3, line 93-94: Change to “when first acclimated to a lower salinity” to improve readability.

12. Page 3 lines 95-96: Change to “and these species show that they can rapidly adapt to new environmental conditions after a few generations” to improve readability.

13. Page 3, lines 99-100: Change to “but we note that quagga mussels have not be detected in brackish waters to date.” To improve readability.

14. Page 3, lines 103-104: Change to “have the potential for introduction to the western basin of the United States where there are many saline and freshwater lakes” I am not sure what the “terminal lakes” is referring to in this sentence. It should be clarified as few readers will understand what it is a reference to.

15. Page 3, line 104: Change to “we conducted site specific studies” to improve grammar.

16. Page 3, line 106: Change “is” to “are” to match pleural noun “freshwaters”.

17. Page 3, line 109: Change to “the objectives were to” to stay in properly in past tense.

18. Page 3, lines 110, 113, and 114: Change to “determine”, “explore” and “analyze” to properly stay in lower case within this single sentence.

19. Page 4, line 120: Change to “semi-natural”.

20. Page 4, line 121: Change to “in water from two”.

21. Page 4, line 123: Change to “which were strategically”.

22. Page 4, line 131: Change to “Sierra Nevada Mountains”.

23. Page 4, line 132: Change to “located on the Colorado River”.

24. Page 4, line 138-139: Change to “conditions quagga mussels”.

25. Page 4, lines 146-147: Change to “at the University of California Analytical Laboratory” so that readers know exactly where the testing took place and its name should be capitalized because it is a specific place.

26. Page 4, line 148: Change to “at the High Sierra Water Laboratory” to give the full name of the institution and improve readability.

27. Page 5, lines 161-162: Change “and the mass” to “and their mass” to improve readability.

28. Page 5, line 163: Change to “Then we placed” to stay in past tense.

29. Page 5, line 165: Change to ‘(details below)”.

30. Page 5, line 176: Change to “Experimental Design” to improve grammar.

31. Page 5, line 180: Change to “mussels were able” to stay in past tense.

32. Page 5, line 187: Change to “survival in Pyramid Lake water,” to improve grammar.

33. Page 5, line 190: Change to “small size in each”.

34. Page 5, line 193: Change to “adult quagga mussel were able to survive and grow in the Truckee River” to stay in past tense and improve readability.

35. Page 5, line 195: Change to “Lake Mead water as a control” to improve grammar.

36. Page 6, line 203: Change to “and shipped it to” to improve grammar.

37. Page 6, line 219: Change to “growth of quagga” to improve readability.

38. Page 6, line 232: Change to “increasing pH reduced the survival” to stay in past tense.

39. Page 6, line 235: Change to “the following levels: 9.0, 9.3, 9.5, 10, and 11” because pH is not measured in terms of concentration”.

40. Page 6, line 237: Change to “Thirty adult mussels” because a sentence should not be started with a number.

41. Page 7, lines 243 and 244: Insert a space between these two lines to match that used throughout the manuscript.

42. Page 7 line 244: Change to “Data were” to stay in past tense.

43. Page 7, lines 246-247: Change sentence to “Linear regression was used to analyze mortality and growth rates in relation to size.”, to improve readability.

44. Page 7, lines 249-250: Change sentence to “Differences in mortality rates between Truckee River and Lake Mead mussels under different pH treatments were tested using a LMM with site and pH modeled as fixed effects.” To improve readability.

45. Page 7, line 262: Change to “The two Pyramid Lake locations were” to improve readability.

46. Page 7, line 265: Change to “The highest chlorophyll a concentrations, utilized” to improve accuracy of the statement.

47. Page 7, line 270: Change to “9.3 at both sites” to improve readability.

48. Page 7, line 271: Insert a coma after “locations” to improve grammar.

49. Page 7, line 272: Change “Dissolve” to “Dissolved” to improve grammar.

50. Page 7, line 275: Change to “Sodium (Na) and Ca were the dominant” to stay in past tense.

51. Page 7, lines 276-277: Change to “with concentrations of 40.0 ± 3.4 mg L-1 and 16.0 ± 1.2 mg L-1, respectively.” to improve readability.

52. Page 8, line 286: Change to “In the Warrior point and Popcorn Rock treatments” to improve grammar.

53. Page 8, Line 297: Change to “Mussel closing of the shell gape in response to touch was very slow before death.” To improve readability.

54. Page 8, line 303: Change to “Quagga mussels can survive” (i.e., remove “The”) to improve readability.

55. Page 9, line 347: Change to “in the Mead and Truckee water treatments” to improve grammar and readability”.

56. Page 9, line 351-352: Change to “Only one mussel died after 14 days at pH 9.0 in Lake Mead water and at pH 9.0 and 9.3 in Truckee River water (Figure 5).” To improve readability.

57. Page 9, line 354: Change to “in Lake Mead water treatments at pH 9.5, 10, and 11 “ to improve readability.

58. Page 10, line 363: Change to “Our results showed that adult” to stay in past tense.

59. Page 10, lines 374-375: Change to “The combined results indicated that” to improve grammar and stay in past tense.

60. Page 10, lines 376-377: Change to “Our study lacks analysis of the capacity of quagga mussel veligers to complete their life cycle in the Lower Truckee River, thus ensuring a permanent population.” to improve readability.

61. Page 10, line 381: Change to “The Truckee River presents” to improve grammar.

62. Page 10, line 388: Change to “a low risk of quagga mussel invasion considering” to improve grammar.

63. Page 11, lines 401-402: Change to “to be inappropriate since there is a knowledge gap regarding this species’ environmental tolerances.”

64. Page 11, line 408: Change to “in both Pyramid Lake sites”

65. Page 11, line 410: Change to “the rapid mortality rate”.

66. Page 11, line 411-412: Change to “that the maximum salinity tolerance”.

67. Page 11, line 415: change to “Acclimatization to a prior salinity of 2 ppt” to improve readability.

68. Page 11, line 421: Change to “damage in adult quagga mussels exposed”.

69. Page 11, lines 431-432: Change to “Survival of zebra mussels in higher salinities depends” to improve readability

70. Page 11, line 433: Change “gain” to gained”.

72. Page 11, line 433 and 434: Change to “A low concentration”.

73. Page 11, lines 431-438: Since this section refers only to studies with zebra mussels that should be made clear. Suggest rewriting as:
“Dreissena species are sensitive to ionic concentrations. Survival of zebra mussels in higher salinities depends on the balance of potassium (K) and Na that allows them to eliminate water gained over an elevated osmotic gradient (Dietz & Byrne, 1997). Low concentrations of K can kill zebra mussels within 6-24 h by inhibiting respiration and reducing filtration rates (Fisher et al., 1991; Horohov et al., 1992). Zebra mussel veligers cannot settle and adults cannot attach to substrate at concentrations of K as low as 10-30 mg L-1 (Fisher et al., 1991) and their valve closure is inhibited at K concentrations of about 70-80 mg L-1 (Wildridge et al., 1998).”

74. Pages 11-12, lines 439-441: Change to “The slow reaction to touch showed by quagga mussels in our experiments when testing for survival could result from stress induced by high concentrations of potassium.” To improve readability.

75. Page 12, line 441: Change to “Also, zebra mussels” to clarify that the research being referred to was on zebra mussels.

76. Page 12, line 445: Change to “sense and respond to toxins by closing their shell valves closed for periods greater than 48 h” to improve readability.

77. Page 12, line 460: Change to “facilitated establishment of quagga mussel populations under very variable” to improve readability.

78. Page 12, line 462: Change to “Dreissena” to the more appropriate “Dreissenid”.

79. Page 12, line 464: Change to “Dreissena” to the more appropriate “Dreissenid”.

80. Page 12, line 464: Change to “In addition, studies of zebra and quagga mussel tolerances were made”.

81. Page 12, line 472: Change to “their tolerance to water body physical and chemical conditions.” To improve readability.

82. Page 12, line 473: Change to “potential for quagga mussel” to improve readability.

83. Page 12, line 474: Change to “manipulating salinity” to improve grammar.

84. Page 13, line 480: Change to “In Pyramid Lake, decrease in pH and temperature” to improve grammar.

85. Page 13, line 486: Change to “except at pH>10.” to improve readability.

86. Page 13, Line 490: Change to “the upper tolerance limit of adult quagga mussels.” to improve readability.

87. Page 13, lines 494-495: Change to “contrasted with the possibility of quagga mussel invasion of higher salinity water bodies, suggests the need for further investigation of the impacts of water physical and chemical conditions on their invasion potential, especially in regard to salinity.” to improve readability.

88. Page 13, line 497: Change to “potential for quagga mussel invasion,” to improve readability.

89. Page 13, line 506: Change to “invaluable in their knowledge” to improve readability.

90. Page 14, line 537: Change to quagga mussels.

91. Page14, line 543: Change polymorphia to polymorpha.

92. Page 14, line 545: Change McMahon R to McMahon RF.

93. Page 14, lines 556-557: Use lower case on article title.

94. Page 15, line 572: Change Dereissena to Dreissena.

95. Page 15, line 585-586: Change article title to “The physiological ecology of the zebra mussel, Dreissena polymorpha, in North America and Europe.

96. Page 15-16, lines 612-613: Article title should not be capitalized.

Annotated reviews are not available for download in order to protect the identity of reviewers who chose to remain anonymous.

---

## Round 0.2 · Minor Revisions

Thank you very much for your thorough revision which addressed almost all concerns of the reviewers. Now, we are almost there with only a few editorial and grammatical suggestions to be solved, as proposed by reviewer 3. I look forward to your final revision.

Reviewer 1 ·

Basic reporting

The revised manuscript is significantly improved, addressing issues of grammar and clarity in the initial submission.

Experimental design

no comment

Validity of the findings

no comment

Reviewer 3 ·

Basic reporting

The authors’ revisions to the manuscript have greatly improved it. The authors appear to have positively responded to this reviewer’s and the other reviewers suggestions for revision and improvement. The revised manuscript to be very readable with the Methods and Results sections clear and understandable and the Discussion section describing the importance of the results relative to previous studies of the environmental tolerances of dreissenid mussels, particularly with regard to pH and salinity. It was important that the study was of the pH and salinity tolerances of quagga mussels because the majority of US dreissenid environmental tolerance studies in this area have been conducted on zebra mussels with only limited information available for quagga mussels.

Experimental design

The experimental design, analysis of results, discussion and citations all appeared appropriate. Statistical approaches to data analysis appeared sound. Figures 1-6 and Table 1 supported the information presented and were easily understood.

Validity of the findings

The results of the study indicated that specimens of the quagga mussel Dreissena rostriformis bugensis could not tolerate the high salinity (4 ppt) of Pyramid Lake. Mussels also experienced high levels of mortality when exposed to Pyramid Lake water modified to have pH levels below those tolerated by quagga mussels in Lake Mead which has dense, sustainable mussel population. These results suggested the mussel mortality in Pyramid Lake water was due to factors of than high pH, specifically high salinity and potentially elevated concentrations of other ions. The pH of Pyramid Lake water was 9.3. Mussels from Lake Mead and Truckee River with a pH of 8.3, and 8.1, respectively, were able to survive exposure to a pH of 9.5 which again indicated that high pH was not the cause of mussel mortality when specimens were exposed to Pyramid Lake water.

The study’s results indicated that quagga mussels were likely to be able to establish a sustainable population if introduced to the Truckee River with salinity levels similar to those of Lake Mead from which it flows, while the elevated saline conditions of Pyramid Lake into which the Truckee River flows were likely to make it highly resistant to establishment of a sustainable quagga mussel infestation. Such information is important for development of risk assessments for quagga mussel invasion of southwestern water bodies where there are a number of saline terminal lakes. Thus, the manuscript should be of interest to readers of PeerJ, especially those working in area of aquatic invasive species and particularly those involved with the developing water body dreissenid risk assessments in the US and the dispersal and monitoring of dreissenid mussel invasion of North American water bodies especially for quagga mussels for which there have been few studies of environmental resistance compared to zebra mussels.

Additional comments

Because the authors responded positively to this reviewer’s suggestions for improvement of the manuscript, the reviewer had no major concerns or suggestions for further major revision. However, on reading the revised manuscript, it was found that some further minor corrections and improvements were required. These suggestions for improvement and correction can be found in the “Minor Editorial and Grammatical Suggestions” and the “Minor Corrections to Literature Cited” sections of the review detailed below.

Minor Editorial and Grammatical Suggestions

1. Line 25: Change to “invasion using specific information from quagga mussel life history or experiments that test” (remove second “from” from the line).

2. Line 26: Suggest changing to “for their survival in the fresh and saline waters of the western United States.

3. Line 95: Suggest changing to “intensively examined for zebra mussels than quagga mussels. Salinity tolerances have been” to improve readability and grammar.

4. Lines 95-97: Suggest changing this sentence to “Salinity tolerances determined by laboratory experiments and presence-absence data are highly variable, but suggest that adult mussels can tolerate up to 6-8 ppt when first acclimated to a lower salinity concentration (Garton, McMahon & Stoeckmann, 2014).” To improve readability.

5. Line 248: Suggest removing extra space between “a” and “on” in line 248, i.e., the sentence “pheophytin-corrected chlorophyll a on a Turner 10-AU fluorometer following methanol”.

6. Lines 377-378: Suggest rewriting these lines as “in this lake with the relationships between growth and shell length indicative of stressful conditions for mussels (Claudi et al., 2013).” to improve sentence readability.

7. Line 453: Remove extra space between “126 mg L-1 and (Reddy & Hoch, 2021)”

8. Caption for Fig. 3. The caption for this Fig 3 was incomplete in materials sent to the reviewer, i.e. “Linear regression between the difference in length and weight between the start and the end of the experiments in the Lake Mead with numbers indicating individual mussels in the experiment (A) and Truckee River (B) treatments. Barplot showing the sex and---? Should this missing part of the figure caption be “sex and gonadal condition”? Also remove “the” from “in the Lake Mead” to improve readability.


Minor Corrections to Literature Cited

1. Line 547: Change to “Baldwin BS, Carpenter M, Rury K., Woodward E. 2012”.

2. Line 553: Italicize Dreissena spp.

3. Line 565: Suggest a change to “Andrew N. Cohen. 2007. Potential Distribution of Zebra Mussels (Dreissena polymorpha) and Quagga Mussels (Dreissena bugensis) in California, Phase 1 Report for the California Department of Fish and Game. San Francisco Estuary Institute, Oakland, California.”

4. Line 584: Change “im pacts” to “impacts”.

5. Line 586: Change (Cerastoderma edule) to (Cerastoderma edule).

6. Line 609: Italicize Dreissena polymorpha and Dreissena rostriformis bugensis.

7. Line 629: Change “Ernie May B.” to “May B.”

8. Line 633: Change “An Overview” to “An overview”.

9. Line 639: Change “Wildland Fire Operations” to “wildland fire operations”.

10. Lines 646-651: These two references (i.e. Fantle-Lepczyk JE, Haubrock PJ, Kramer AM, Cuthbert RN, Turbelin AJ, Crystal-Ornelas R, Diagne C, Courchamp F. 2022. and Horohov J, Silverman H, Lynn JW, Dietz TH. 1992. Are alphabetically out of place in the list of references. Both references are correctly cited on lines 576-578 and 593-595, respectively, so these repeated references in lines 646-652 should be eliminated.

11. Lines 649-651: This reference to “Ram JL, Karim AS, Banno F, Kashian DR. 2012. Invading the invaders: reproductive and other mechanisms mediating the displacement of zebra mussels by quagga mussels. Invertebrate Reproduction & Development 56:21–32. DOI: 10.1080/07924259.2011.588015.” is alphabetically out of place and redundant. It properly appears in lines 661-663 so this citation to it in lines 649-651 should be removed from the References Section.

12. Lines 655-656; Is the reference “Pinheiro J, Bates D, DebRoy S, Sarkar D, Team. RC. 2020. nlme: Linear and Nonlinear Mixed Effects Models.” correct? A search of the web indicated that it might be “Pinheiro J, Bates D, R-core. 2020. Nime: Linear and Nonlinear Mixed Effects Models.”

13. Lines 657-659. This reference could be improved if written as “Pucherelli S, O’Meara S, Kevin B, Kirsch J. 2016. Habitat Suitability Parameters for Quagga Mussels in the Lower Colorado River System and at Reclamation Managed Facilities. Research and Development Office, Bureau of Reclamation, U.S. Department of the Interior, Denver CO.

14. Line 689: Change “united states” to United States.

15. Line 695: Italicize Dreissena bugensis and Dreissena polymorpha.

Annotated reviews are not available for download in order to protect the identity of reviewers who chose to remain anonymous.

---

## Round 0.3 · accepted · Accept

Thank you very much for the thorough revision. Now, all remaining comments of reviewer 3 are fully addressed as assessed by myself so that I am happy with the current version. Your manuscript is now ready for publication.